# Antioxidant Activity Analysis of Native *Actinidia arguta* Cultivars

**DOI:** 10.3390/ijms25031505

**Published:** 2024-01-25

**Authors:** Yu Kyong Hu, Soo Jae Kim, Cheol Seong Jang, Sung Don Lim

**Affiliations:** 1Molecular Plant Physiology Laboratory, Department of Applied Plant Sciences, Graduate School, Sangji University, Wonju 26339, Republic of Korea; 2022011704@sj.sangji.ac.kr; 2Wonju-si Agricultural Technology Center, Heungdae-gil 7, Heungup-myeon, Wonju 26339, Republic of Korea; idnature@korea.kr; 3Plant Genomics Laboratory, Interdisciplinary Program in Smart Agriculture, Kangwon National University, Chuncheon 24341, Republic of Korea

**Keywords:** flavonoids, fruit, kiwiberry, phenolics, phytochemical content, sugar content, vegetative tissues, vitamin C

## Abstract

Kiwiberry (*Actinidia arguta*) is a perennial fruit tree belonging to the family Actinidiaceae. Kiwiberries are known to have an extremely high concentration of sugars, phenolics, flavonoids, and vitamin C, and possess delicious taste and health-promoting properties. Numerous studies have focused on kiwiberry fruits, demonstrating that they possess a higher phytochemical content and greater antioxidant activities than other berry fruits. The purpose of this study was to compare the phytochemical content and antioxidant potential of leaf, stem, root, and fruit extracts from twelve kiwiberry cultivars grown in Wonju, Korea, characterized by a Dwa climate (Köppen climate classification). In most kiwiberry cultivars, the total phenolic (TPC) and total flavonoid (TFC) phytochemical content was significantly higher in leaf and stem tissues, while the roots exhibited higher antioxidant activity. In fruit tissues, the TPC and TFC were higher in unripe and ripe kiwiberry fruits, respectively, and antioxidant activity was generally higher in unripe than ripe fruit across most of the cultivars. Based on our results, among the 12 kiwiberry cultivars, cv. Daebo and cv. Saehan have a significantly higher phytochemical content and antioxidant activity in all of the tissue types, thus having potential as a functional food and natural antioxidant.

## 1. Introduction

Kiwiberry (*Actinidia arguta*) belongs to the family Actinidiaceae, which comprises 75 species, 4 of which, *A. arguta*, *A. rufa*, *A. polygama*, and *A. kolkmikta*, are predominantly distributed in Korea [1,2]. It is a dioecious, climbing deciduous fruit tree that thrives in temperate regions [3]. While kiwifruits (*A. deliciosa*) are sensitive to cold temperatures, kiwiberries have robust cold and disease resistance, which facilitates more sustainable farming practices [4,5]. For this reason, kiwiberries are well adapted to mountainous regions and grow naturally in Korea, China, and Japan [6]. The growth of the kiwiberry is greatly affected by climate, soil, and topographic conditions. Its optimal quality and yield are achieved in cultivation areas with a humid subtropical monsoon climate, with excellent hydrothermal conditions and abundant precipitation throughout the year [7,8]. Furthermore, high altitudes and hot weather often cause significant damage to kiwiberry fruits and negatively affect fruit quality [9]. 

Kiwiberry cultivation began on a small scale in the United States, Chile, and Europe (France and Switzerland) between 1980 and 1990 [10]. Over the past few years, the kiwiberry has gained popularity among producers and consumers in countries such as the USA, New Zealand, Korea, Japan, Chile, and Europe [11]. According to the most recent report, global kiwiberry production in 2015–2016 reached almost 1600 tons, and domestic kiwiberry production in Korea has also shown a notable increase, reaching approximately 130 tons in 2023 [12,13]. This production growth is indicative of the rising popularity and demand for kiwiberries, with the entire Korean yield consumed domestically. Historically, various kiwiberry tissues, including fruits, leaves, and roots, have been used for food and medicinal purposes [14]. In some southern regions of Korea, the young leaves and stems of the kiwiberry are harvested in spring and either boiled as vegetables or dried for consumption during the winter [15]. Additionally, young shoots and leaves are harvested and consumed as tea [16]. The roots are known to have anticancer and anti-inflammatory effects and have been used to treat liver, lung, and myeloma tumors [17].

Fruits are a major dietary source of various antioxidant phytocompounds in humans. The fruits of Actinidia species, as well as citrus fruits, are excellent sources of vitamin C [18]. Kiwiberries also possess a high oxidant capacity, primarily due to their substantial phenolic content, ranking them as the second highest antioxidant fruit after plums [19,20,21]. Although kiwis and kiwiberry contain several different classes of bioactive compounds, their antioxidant properties are predominantly associated with phenolic compounds and ascorbate [22,23]. Phenolics, especially polyphenols, protect against cardiovascular diseases, diabetes, osteoporosis, and neurodegenerative diseases [24]. Phenolic compounds also reduce the risk of various chronic diseases, whereas flavonoids exhibit anti-inflammatory, antiallergic, anticancer, and antiulcer properties [25]. In addition to the possible pro-health properties of these phytocompounds, they are important in the interaction between plants and their environment, such as serving as pollination attractants and feeding deterrents, as well as protecting against various biotic and abiotic stresses owing to their antioxidant activity [26,27]. 

Every part of the kiwiberry plant is edible and enriched with beneficial active ingredients. Each component has unique properties, is influenced by specific environmental conditions, and is essential for plant survival. Considering these properties, the identification of bioactive compounds can help elucidate their specific physiological and biochemical activities and indicate their potential for medicinal applications. Numerous studies have focused on the antioxidant properties of kiwiberry fruits. However, the antioxidant effects of kiwiberry extracts from vegetative tissues and fruits have not been extensively studied. 

Therefore, the objectives of this study were to evaluate the phytochemical compounds contributing to the total phenolic (TPC) and total flavonoid (TFC) content and antioxidant capacity of vegetative tissues and fruits from twelve kiwiberry cultivars: cv. Gwangsan, cv. Green Ball, cv. Green Heart, cv. Daebo, cv. Daesung, cv. Saehan, cv. Apple, cv. Autumn Sense, cv. Wangneujdarae, cv. Cheongsan, cv. Cheongyeon, and cv. Chiak. Comprehensive analysis of both biomass characteristics and antioxidant properties in leaf, stem, root, and fruit tissues of numerous kiwiberry cultivars can provide information on the optimal antioxidant effectiveness for various applications, ranging from dietary supplements to functional food products.

## 2. Results and Discussion

### 2.1. Leaf Biomass and Water Content of Kiwiberry Cultivars

Leaf biomass parameters, including leaf area, fresh weight (FW), dry weight (DW), and water content (WC), were measured across 12 kiwiberry cultivars: Gwangsan, Green Ball, Green Heart, Daebo, Daesung, Saehan, Apple, Autumn Sense, Wangneujdarae, Cheongsan, Cheongyeon, and Chiak (Figure 1). Significant differences were detected among the tested cultivars in leaf biomass parameters. Excluding cv. Daebo, the leaf area of the other 11 cultivars was relatively consistent, ranging from 75.07 ± 2.33 to 81.47 ± 2.85 cm^3^ (Figure 1A,B). The cv. Daebo had the highest leaf area of 86.02 ± 2.44 cm^3^, which is 1.14-fold higher than that of cv. Chiak, with the smallest leaf area. The cv. Daebo also exhibited a significantly higher FW and DW, 2.98 ± 0.46 and 0.63 ± 0.10 g of leaves, respectively (Figure 1C,D). Although cv. Daebo had the highest leaf biomass, it had the lowest leaf WC (70.74 ± 1.98%) compared to the average WC of the other cultivars (Figure 1E). This observation suggests the possibility of denser leaf tissue in cv. Daebo, which is typically linked to enhanced mechanical strength and the potential for drought resistance in plants. In contrast to angiosperms, which typically show an association between drought tolerance and increased mechanical toughness, cycads tend to display greater drought tolerance with thinner and less mechanically tough leaves [28,29,30]. The diverse leaf morphology among the studied kiwiberry cultivars suggests that considerable variations in leaf biomass capacity could influence their potential applications and the selection of the most efficient cultivars for obtaining optimal antioxidant effectiveness.

### 2.2. Total Phenolic and Flavonoid Content in Vegetative Tissues of Kiwiberry Cultivars

The differences in TPC and TFC of extracts from vegetative tissues of 12 kiwiberry cultivars are shown in Figure 2. The TPC of leaves, young stems, and roots ranged from 21.26 ± 0.64 to 60.05 ± 5.04 mg gallic acid equivalent (GAE)·g^−1^, 17.64 ± 0.21 to 41.80 ± 1.17 mg GAE·g^−1^, and 7.26 ± 0.55 to 45.27 ± 1.60 mg GAE·g^−1^, respectively (Figure 2A). The cv. Daebo showed significantly higher TPC in its leaves, being 2.8-fold greater than that of cv. Cheongsan. However, in the stem and root tissues, cv. Daebo exhibited the lowest TPC compared to the other cultivars. The cv. Autumn Sence and the cv. Saehan exhibited a 2.3- and 6.2-fold higher TPC than that of cv. Daebo in their stem and root tissues, respectively. 

The TFC displayed notable variability across the different tissues of the 12 cultivars (Figure 2B). Leaves exhibited a TFC ranging from 31.25 ± 2.43 to 64.43 ± 1.95 mg quercetin equivalent (QE)·g^−1^, while young stems and roots varied from 11.74 ± 2.21 to 20.20 ± 1.26 mg QE·g^−1^, and 7.59 ± 2.06 to 17.46 ± 0.88 mg QE·g^−1^, respectively. In the leaf samples, the cultivar cv. Daesung contained a TFC that was double that of cv. Apple. For the stem tissues, the TFC of cv. Cheongyeon was 1.7-fold higher than that of cv. Chiak. Similarly, in the root samples, the TFC of cv. Saehan was 2.3-fold higher than that of cv. Wangneujdarae. Collectively, analyzing extracts from the leaves, stems, and roots of the 12 kiwiberry cultivars revealed that, in most cases, the TPC and TFC were highest in leaf tissues, followed by stems and then roots. These results are consistent with previous studies, in which the most abundant TPC and TPC were generally found in the leaves of natural plants rather than in bred cultivars [31,32]. For example, a study on the leaves, stems, and roots of baekseohyang (*Daphne kiusiana*) indicated that the highest TPC (24.39 ± 1.21 mg GAE·g^−1^) and TFC (9.38 ± 0.87 mg rutin equivalent·g^−1^) and the lowest TPC (12.85 ± 0.42 mg GAE·g^−1^) and TFC (3.11 ± 0.82 mg RE·g^−1^) were detected in the leaf and root tissues, respectively [33]. Analysis of *Achyranthes bidentata*, a plant valued for both its edible and medicinal properties, revealed that the TPC of the leaf extract was 58.27 ± 0.75 mg GAE·g^−1^, while stem extracts had a TPC of 16.13 ± 0.42 mg GAE·g^−1^ [34,35]. In comparison, the TPC in the leaves and stems of kiwiberry cultivars was found to be 4- and 2-fold higher, respectively, than those in widely recognized medicinal plants and functional foods, such as *Angelica gigas*, *Morus alba*, and red ginseng [36]. However, the methanol extract of *Actinidia kolkmikta* roots demonstrated a TPC of 201.9 mg GAE·g^−1^, which is 20-fold higher than that observed in *A. arguta* in our study, suggesting that different extraction methods potentially influence the concentrations detected [37]. Our study was the first to compare the TPC and TFC in vegetative tissues of different kiwiberry cultivars and found that they all contain high levels of TPC and TFC, suggesting significant potential for their use in functional and medicinal products.

### 2.3. Antioxidant Activity in Vegetative Tissues of Kiwiberry Cultivars

The antioxidant activities of the leaf, stem, and root extracts from the 12 kiwiberry cultivars are shown in Figure 3. The 2,2-diphenyl-1-picrylhydrazyl (DPPH) and 2,2′-azino-bis (3-ethylbenzothiazoline-6-sulfonic acid) (ABTS) radical scavenging activities (RSAs) varied significantly among the different vegetative tissues of the 12 cultivars. The average values of DPPH RSAs from the 12 kiwiberry cultivars were highest in the root (75.80 ± 14.35%), followed by the leaves (53.61 ± 14.90%), and then the stems (34.86 ± 17.05%) (Figure 3A). Similarly, for ABTS RSAs, the root extract exhibited the highest average values at 92.8 ± 0.42%, compared to the leaf (57.95 ± 18.9%) and stem extracts (30.7 ± 20.09%) (Figure 3B).

When comparing cultivars, the leaf extract from cv. Daebo had a DPPH RSA of 75 ± 7.74%, which was 2.5-fold higher than that of cv. Cheongsan at 29 ± 2.14% (Figure 3A). In cv. Saehan, the stem and root extracts exhibited DPPH activities of 67 ± 2.16% and 86 ± 0.44%, which were 3.5-fold and 2.2-fold higher than those of cv. Green Ball, respectively, while the root extract had a 3.6-fold higher DPPH activity than that of cv. Daebo (Figure 3A). Regarding the ABTS RSA in different tissues from the 12 cultivars, the leaf extract from cv. Daebo exhibited 87 ± 0.99% activity, which was 3.6-fold higher than that of cv. Cheongsan (Figure 3B). The young stems of cv. Saehan showed 67 ±3.47% ABTS activity, which is 4.9-fold higher than that of cv. Wangneujdarae. In roots, cv. Chiak exhibited a high value of 93 ± 0.12% ABTS activity, but there was no significant difference among cultivars. The ABTS RSA removes artificial radicals in the same way as the DPPH RSA, and the two have been reported to be significantly correlated [38]. For this reason, both the DPPH and ABTS RSA exhibited similar patterns in the extracts from different tissues of kiwiberry cultivars (Figure 3A,B).

The average nitrite scavenging activity (NSA) of the leaf, stem, and root extracts from kiwiberry cultivars was 96 ± 0.17, 98 ± 0.21, and 89 ± 1.16%, respectively, with no significant difference observed between leaves and stems among the cultivars. However, in the roots, the NSA in cv. Saehan was 1.2-fold higher than that in cv. Daebo (Figure 3C). Unlike the DPPH and ABTS RSAs, which were higher in the root tissues, the NSA was lower in the root than the leaf and stem tissues in most kiwiberry cultivars. The NSA of these kiwiberry cultivars is significantly higher compared to medicinal plants like *Artemisia princeps* and *Achyranthes bidentata* [35,39]. Specifically, *Artemisia princeps* demonstrated an NSA of 41 ± 0.70% in leaves, and *Achyranthes bidentata* showed 15 ± 0.40% in stems. This comparison highlights the notably stronger nitrate scavenging capacity of kiwiberry cultivars.

The reducing power of a compound can serve as a significant indicator of its antioxidant activity. Antioxidants donate electrons to reactive radicals, reducing them to more stable and unreactive species [40]. The average values of the reducing power of the 12 kiwiberry cultivars were highest in the root (1.46 ± 0.64 at OD_700_), followed by the leaves (0.66 ± 0.21 at OD_700_), and then the stems (0.59 ± 0.22 at OD_700_). However, the difference between leaves and stems was not significant (Figure 3D). The ability of the root extracts to react with free radicals was in the following order: kiwiberry cv. Saehan (3.3 ± 0.44) > Chiak (2.01 ± 0.27) > Daesung (1.65 ± 0.23) > Cheongyeon (1.58 ± 0.07) > Gwangsan (1.55 ± 0.21) > Autumn Sense (1.29 ± 0.04) > Cheongsan (1.21 ± 0.08) > Apple (1.19 ± 0.05) > Green Heart (1.04 ± 0.05) > Green Ball (0.95 ± 0.02) > Daebo (0.89 ± 0.03) > Wangneujdarae (0.86 ± 0.04). Notably, the highest value observed in cv. Saehan was 3.7-fold greater than the lowest value in cv. Wangneujdarae. The reducing power of *Chrysanthemum zawadskii* was found to be 0.96 and 1.02 in leaf and stem tissues, respectively [39]. In addition, barley sprouts showed a reducing power of 0.23 in leaves, 0.12 in stems, and 0.18 in roots [40]. Our results thus indicate that kiwiberries have a significantly higher reducing power than those other plants.

Collectively, the results indicate that regardless of the kiwiberry cultivar, the antioxidant activities of DPPH and ABTS scavenging activity and reducing power are significantly enriched in the root tissues compared to the leaf or stem tissues. However, the NSA was higher in the leaf and stem tissues in most kiwiberry cultivars (Figure 3). DPPH and ABTS are widely recognized as effective methods for studying and evaluating the free RSA and antioxidant activity in plants [41]. Additionally, the antioxidant activity in leaves can be linked to the presence of phytochemicals such as phenolic and flavonoid compounds, which are known to act as natural antioxidants [42]. In the present study, most kiwiberry cultivars contained higher TPC and TFC in leaf tissues, whereas the DPPH and ABTS RSA and reducing power were higher in root tissues (Figure 2 and Figure 3). The antioxidant activities in different plant tissues can be significantly influenced by the type and structure of the active compounds present. The presence of specific compounds [43] in root tissues of the kiwiberry, such as terpenoids (e.g., 2α,3β-dihydroxyurs-12-en-28,30-olide, 12α-chloro-2α,3 β,23-tetrahydroxyolean-28-oic acid-13-lactone, and oleanolic acid) [44,45,46], flavonoids (e.g., (−)-epi-catechin, (+)-catechin, and Procyanidin B4) [47], phenolics (planchol A, B and isotachioside) [44], and other unique organic and volatile compounds, can explain the higher antioxidant activity in the roots compared to the leaves [48]. For example, oleanolic acid is a natural terpenoid known to function as a powerful antioxidant to scavenge free radicals and protect against oxidative stress [49,50]. Flavonoids like catechins are also well-known antioxidants [51,52]. This unique combination of bioactive compounds in the roots likely contributes to the higher antioxidant activity compared to other natural and medicinal plants [53].

### 2.4. Fruit Biomass, Sugar, and Vitamin C Content of Kiwiberry Cultivars

After fertilization, fruit development is dependent on both the processes of cell division and cell expansion [54,55]. To evaluate fruit biomass, FW, DW, and WC of unripe (60 days after pollination: DAP) and ripe (90 DAP) kiwiberry fruits were measured and compared between the 12 cultivars (Figure 4A–C). The variation between FW and DW across different cultivars was more pronounced in ripe than unripe kiwiberry fruits (Figure 4A,B). The FW of unripe or ripe kiwiberry fruits from cv. Daebo was 1.3 (3.37 ± 0.70 g)- and 2.5 (29.03 ± 2.41 g)-fold higher, respectively, compared to the average FW of unripe (2.54 ± 0.36 g) or ripe (11.43 ± 4.1 g) fruits from the other 11 cultivars (Figure 4B). Among the kiwiberry cultivars, cv. Daebo and cv. Saehan had the highest average DW of ripe fruits at 3.29 ± 0.39 and 2.24 ± 0.16 g, respectively. In contrast, cv. Wangneujdarae and cv. Apple had a significantly lower average DW of 1.04 ± 0.12 and 1.18 ± 0.10 g, respectively, in their ripe fruits (Figure 4C). The average WC of unripe and ripe kiwiberry fruits across cultivars was 86.6 ± 2.30 and 83.2 ± 2.93%, respectively. The unripe cv. Gwangsan (89.03 ± 1.38%) showed a WC that was 1.08-fold higher than that of cv. Autumn Sense (81.91 ± 4.43%). Similarly, cv. Daebo (89.01 ± 1.37%) exhibited a WC 1.15-fold higher than that of cv. Cheongyeon (78.08 ± 1.93%) in ripe fruits (Figure 4D). Jeong et al. [56] reported that domestic gold kiwifruit (*A. chinensis*) has an average moisture content of 78.62%. Comparatively, the WC of kiwiberry cultivars in this study tended to exceed that of gold kiwifruit. When comparing the biomass of unripe and ripe kiwiberries, both the FW and DW increased in ripe fruits (Figure 4B,C), although unripe kiwiberries have a relatively higher moisture content (Figure 4D).

The sugar and vitamin C content of ripe fruits of the 12 kiwiberry cultivars were evaluated in Figure 4E,F. The measurement of sugar content using both a refractometer and carbohydrate assay showed a very similar trend. The sugar content of ripe kiwifruits ranged from 956.77 ± 144.57 mg·100 g^−1^ (8.66 ± 0.66 °Bx) in cv. Daebo to 2102.05 ± 327.51 mg·100 g^−1^ (16.33 ± 1.41 °Bx) in cv. Wangneujdarae, and the average sugar content of the 12 cultivars was 1474.255 ± 326.41 mg·100 g^−1^ (12.24 ± 2.34 °Bx). The ripening of kiwifruits involves the conversion of starch into soluble sugars [57]. Previous studies reported that total soluble sugar content varies greatly between *Actinidia deliciosa*, *A. chinensis*, *A. arguta*, and *A. rufa*. Glucose, fructose, and sucrose are the main sugars in kiwifruit, with *A. arguta* having higher sucrose and myo-inositol levels [58,59,60]. Research on 17 native *A. arguta* varieties showed a substantial sugar content, ranging from 8.20 to 16.57 °Bx, averaging 10.11 °Bx [61]. Although kiwiberry cv. Daebo showed a lower °Bx value, indicating lower sugar due to its higher moisture content, all 12 kiwiberry cultivars studied likely have a higher sugar content than other native kiwiberries (Figure 4E,F).

Vitamin C (ascorbic acid) is recognized as the most important nutrient in kiwiberries. The vitamin C content in ripe kiwiberry cultivars declined in the following order: cv. Cheongsan (208.06 ± 3.24 mg·100 g^−1^) > cv. Autumn Sense (203.41 ± 2.49 mg·100 g^−1^) > cv. Saehan (166.68 ± 1.27 mg·100 g^−1^) > cv. Apple (156.77 ± 0.68 mg·100 g^−1^) > cv. Cheongyeon (152.68 ± 1.23 mg·100 g^−1^) > cv. Wangneujdarae (145.41 ± 7.26 mg·100 g^−1^) > cv. Green Ball (140.35 ± 1.53 mg·100 g^−1^) > cv. Green Heart (134.35 ± 0.33 mg·100 g^−1^) > cv. Daebo (131.02 ± 4.23 mg·100 g^−1^) > cv. Chiak (115.22 ± 1.56 mg·100 g^−1^) > cv. Gwangsan (110.41 ± 0.79 mg·100 g^−1^) > cv. Daesung (91.45 ± 0.23 mg·100 g^−1^) (Figure 4G). The average vitamin C content of the 12 cultivars was 146.32 ± 33.31 mg·100 g^−1^. The cv. Cheongsan and the cv. Autumn Sense showed the highest vitamin C content, 1.42- and 1.39-fold greater compared to the average vitamin C content of all 12 cultivars. 

In commercial cultivars, the vitamin C content of *A. deliciosa*, *A. chinensis*, and *A. arguta* kiwiberry fruits is reported to be within the ranges of 50–250, 50–420, and 80–430 mg·100 g^−1^, respectively [59,62,63]. The vitamin C content of the tested kiwiberry cultivars varied from 91.45 ± 0.23 to 208.06 ± 3.24 mg·100 g^−1^, which was much higher than that of the most commercialized *A. deliciosa* cv. Hayward (59.32 ± 1.89 mg·100 g^−1^) and *A. chinensis* cv. Hort16A (75.1 ± 17.0 mg·100 g^−1^) [18,64]. This high level of vitamin C in kiwiberry fruits might be associated with their high sugar content, because _D_-glucose-6-phosphate derived from sugars is used in the ascorbate synthesis pathway as a precursor [65,66]. Our results suggest that most kiwiberry cultivars contain sufficient vitamin C content in their fruits for adequate human nutrition, in accordance with the European Union’s recommended daily allowance for vitamin C (95 to 110 mg) [67].

### 2.5. Total Phenolic and Flavonoid Content in Fruits of Kiwiberry Cultivars

Phenolic and flavonoid compounds are known to be the main bioactive components in kiwiberry fruits [68,69,70]. To evaluate the TPC and TFC in unripe and ripe kiwiberry fruits, fruit extracts from each of 12 cultivars were analyzed using gallic acid and quercetin as the reference standards (Figure 5). The TPC of unripe fruits varied between 4.05 ± 0.04 mg GAE·g^−1^ in cv. Gwangsan and 9.26 ± 1.04 mg GAE·g^−1^ in cv. Saehan. For ripe fruits, the range was from 3.28 ± 0.10 mg GAE·g^−1^ in cv. Wangneujdarae to 6.23 ± 0.51 mg GAE·g^−1^ in cv. Gwangsan (Figure 5A). The average TPC of the 12 cultivars was higher in unripe fruits, at 6.19 ± 1.51 mg GAE·g^−1^, compared to ripe fruits, which had a TPC of 4.70 ± 0.85 mg GAE·g^−1^. These results indicate that kiwiberry fruits tend to have higher phenolic content in their unripe stage than when ripe. The TPC of developed *A. arguta* cv. Saehan, cv. Green Heart, cv. Cheongsan, and cv. Daesung were 9.26 ± 0.14, 7.94 ± 0.2, 7.32 ± 0.17, and 7.20 ± 0.09 mg GAE·g^−1^, respectively, which was significantly higher than that of the most commercialized *A. deliciosa* cv. Hayward (3.75 ± 0.09 mg GAE·g^−1^) and *A. chinensis* cv. Hort16A (6.60 ± 0.14 mg GAE·g^−1^) [25].

For the TFC in ripe kiwiberry fruits, cv. Daesung showed the highest value at 77.61 ± 10.52 mg QE·g^−1^, while cv. Autumn Sense had the lowest at 60.43 ± 13.10 mg QE·g^−1^ (Figure 5B). In unripe fruits, cv. Cheongyeon presented the highest TFC, at 39.76 ± 1.04 mg QE·g^−1^, and cv. Autumn Sense had the lowest, at 22.94 ± 1.55 mg QE·g^−1^. Contrary to the trend observed with TPC, the TFC in ripe kiwiberry fruits showed higher average values of 69.17 ± 5.53 mg QE·g^−1^ compared to 30.06 ± 4.81 mg QE·g^−1^ in unripe fruits across the 12 cultivars (Figure 5). Overall, this indicates that the TFC is approximately 2.3-fold higher in ripe than in unripe kiwiberry fruits. Previous studies have indicated that both TPC and TFC in fruits exhibit variations between unripe and ripe stages, changing as the fruits reach maturity [71,72]. Specifically, phenolic compound content in unripe fruits is usually higher than in mature fruits of grapes, apples, pomegranates, and kiwifruits [73,74,75,76]. In the case of *A. setona* kiwiberry cv. No.9, cv. CH3, and cv. CH4 fruits, the TPC was approximately 20% higher at 100 days after anthesis compared to 150 days after anthesis [77]. Similarly, *A. deliciosa* cultivars (cv. Abbot, cv. Bruno, cv. Allison, cv. Hayward, cv. Monty, and cv. Jinkui), *A. chinensis* cv. Hongyang, and *A. eriantha* cv. Ganmi No. 6 also exhibited a decrease in TPC as the fruits ripened. Conversely, the TFC was notably lower at the beginning of fruit development, gradually increasing as the fruits matured [78,79]. Phenolic compounds, serving as vital antioxidants and stress-resilient substances, are found in high concentrations in unripe fruits. Their bitter taste may act as a defense mechanism against animal consumption. As the fruits grow and ripen, the phenolic content in both the peel and pulp decreases, reducing astringency. The mechanism leading to the decrease in TPC in fruits as they mature could be attributed to either a decrease in phenolic synthesis or the conversion of phenols into other compounds during fruit development. Additionally, as the fruit weight increases, the concentration of phenolics per unit mass tends to decrease [80,81].

In the present study, we observed that all kiwiberry cultivars exhibited significantly higher TFC in ripe than unripe fruits (Figure 5B). This trend is similar to observations in *A. deliciosa* cv. Abbot, cv. Bruno, cv. Allison, cv. Hayward, and cv. Monty, *Actinidia eriantha* cv. Bidan, *A. arguta* cv. Chiak, cv. Darae No. 2, and *A. chinensis* cv. Haegeum and cv. Haehyang, where ripe fruits have significantly higher TFC levels compared to their unripe fruits [78]. The increase in flavonoid content in ripe compared to unripe fruits can be linked to various factors. The increased TFC in ripe kiwiberry fruits, as a response to unfavorable conditions like spoilage and deterioration, might be explained by the spontaneous activation of its phenol–propan metabolic pathway during maturity [82]. Additionally, as fruits ripen, there is an increase in pigmentation due to the biosynthesis of compounds like anthocyanins, a type of flavonoid (Figure 4A). Attributed to the higher TPC and TFC in kiwiberry cultivars, its nutrient-rich peel and smooth edible skin are a richer source of various phytochemicals for consumption.

### 2.6. Antioxidant Activity in Fruits of Kiwiberry Cultivars

The antioxidant activities of unripe and ripe kiwiberry fruits from 12 cultivars were analyzed using DPPH, ABTS, and NSA assays, along with reducing power ability (Figure 6). Each cultivar showed different levels of antioxidant ability depending on the maturity stage. In the DPPH assay, the antioxidant capacity of the unripe kiwiberry cultivars ranged from 33.92 ± 7.93% (cv. Daebo) to 76.08 ± 1.16% (cv. Green Heart), whereas for the ripe kiwiberries, it ranged from 19.47 ± 2.77% (cv. Chiak) to 35.46 ± 1.14% (cv. Daebo), as shown in Figure 6A. In the ripe fruits, three cultivars, cv. Green Heart (76.08 ± 1.16%), cv. Saehan (73.61 ± 4.14%), and cv. Autumn Sense (66.90 ± 8.57%), exhibited the highest DPPH activity, while cv. Daebo showed the highest DPPH activity among the unripe fruits. The average DPPH activity in unripe cultivars was 53.63 ± 12.84%, which is higher than the 24.92 ± 4.17% activity observed in ripe cultivars. All of the ripe fruits had significantly higher DPPH scavenging activity than the unripe fruits across cultivars. Furthermore, the variance in DPPH activity among different cultivars was not significantly (*p* < 0.05) greater in ripe compared to unripe fruits, except for cv. Daebo, cv. Cheongyeon, and cv. Chiak. This indicates that the differences in antioxidant activity measured by the DPPH assay among cultivars are relatively consistent, regardless of the ripening stage. In the research conducted by Park et al. [78], the DPPH scavenging activity of *Rubus coreanus* was found to be 78.55%. In comparison, a study by Lee et al. [79] reported the DPPH scavenging activities for blueberries and pomegranates as 81.72% and 76.94%, respectively. These values are similar to those observed for unripe kiwiberries but are higher than those recorded for kiwiberries in the post-ripening stage. 

In the ABTS scavenging activity assay, the average antioxidant capacities of each cultivar of unripe kiwiberry fruits greater than those of most ripe fruits, decreased as follows: cv. Green Heart (98.48 ± 0.23%) > cv. Saehan (95.81 ± 0.43%) > cv. Daebo (92.28 ± 0.12%) > cv. Cheongsan (90.51 ± 0.82%) > cv. Daesung (87.09 ± 4.45%) > cv. Autumn Sense (86.23 ± 1.71%) > cv. Wangneujdarae (78.06 ± 0.44%) > cv. Green Ball (63.18 ± 1.73%) > cv. Chiak (61.37 ± 1.12%) > cv. Gwangsan (60.71 ± 1.23%) > cv. Apple (57.23 ± 0.84%) > cv. Cheongyeon (54.05 ± 1.18%) (Figure 6B). The ABTS activity in unripe fruits of six cultivars (cv. Green Heart, cv. Daesung, cv. Saehan, cv. Wangneujdarae, cv. Autumn Sense, cv. Cheongsan, and cv. Chiak) was significantly higher (*p* < 0.05) than in ripe fruits. Similar to the DPPH scavenging activity results, cv. Green Heart exhibited the highest ABTS activity (98.4 ± 0.12%) in unripe kiwiberries, which was 1.8 times greater than that of cv. Cheongyeon. In ripe kiwiberries, cv. Daeseong showed the highest ABTS activity (85.2 ± 1.21%), 1.9 times higher than that of cv. Chiak. When comparing kiwiberry cultivars to other species, strawberries, blueberries, and raspberries have ABTS scavenging activities of 69.83, 99.12, and 65.58%, respectively [83,84]. This comparison suggests that kiwiberries could also be considered a potent natural source of antioxidants. The relatively higher ABTS activity in cv. Green Heart and cv. Saehan correlates with their higher phenolic contents (Figure 5A), indicating a relationship between TPC and ABTS scavenging abilities, similar to the observed trend in DPPH scavenging activity. The variation in antioxidant capacity observed in the tested kiwiberry cultivars can be provisionally attributed to their different TPC and TFC content [85,86].

In the assessment of NSA, a range of antioxidant capacities was observed in kiwiberry fruits (Figure 6C). For unripe fruits, the antioxidant capacity varied from 72.46 ± 3.07% in cv. Cheongyeon to 87.95% ± 0.95% in cv. Saehan. In ripe fruits, this capacity ranged from 60.84 ± 1.88% in cv. Green Heart to 80.21 ± 4.63% in cv. Daebo. The average NSA among cultivars was higher in unripe fruits, with an average of 80.89 ± 3.88%, compared to ripe fruits (71.41 ± 5.67%). This indicates that the unripe kiwiberries possess a strong capability to scavenge nitrites, a property that contributes to their antioxidant potential. The difference in the NSA between unripe and ripe kiwiberries suggests that phenolic components play an important role in scavenging nitrites in kiwiberries (Figure 5A). Similar results have also been reported in other kiwiberry species. In studies examining the NSA of kiwifruits, Chung et al. [58] reported that *A. chinensis* and *A. deliciosa* exhibited 89.87% and 87.68% NSA, respectively. Another study by Chung [59] found that the NSA of black chokeberry (*Aronia melanocarpa*) ranged from 15.1% to 75.7%, while blueberries (*Vaccinium* sect. *Cyanococcus*) ranged from 7.24% to 19.63%. The antioxidant activity in kiwiberry fruits decreased progressively as they matured. This trend can be associated with a reduction in phenol content that typically occurs as the fruit ripens (Figure 5 and Figure 6). Importantly, research on the antioxidant activity of these fruits consistently demonstrated that the phenols present in residual soluble matter from kiwiberry fruits may exhibit robust antioxidant activity. This suggests that the phenolic compounds have the potential to be developed into new antioxidants in kiwiberries [87]. Given that the NSA of kiwiberry is higher than these other fruits, it suggests that kiwiberry extract could be effectively used as a natural functional food material for scavenging nitrites.

The reducing power of unripe and ripe kiwiberries ranged from 0.54 ± 0.01 (cv. Cheongyeon) to 1.43 ± 0.14 (cv. Saehan) at OD_700_ and 0.68 ± 0.03 (cv. Wangneujdarae) to 1.17 ± 0.04 (cv. Daebo) at OD_700_, respectively (Figure 6D). Among them, four specific cultivars showed significant differences (*p* < 0.05) between their ripe and unripe stages. The kiwiberry cv. Daebo (1.17 ± 0.04), cv. Gwangsan (1.13 ± 0.09), cv. Apple (1.08 ± 0.07), and cv. Cheongyeon (0.99 ± 0.07) exhibited a 1.42-, 1.4-, 1.38-, and 1.83-fold higher reducing power in their ripe than their unripe fruits. In contrast, the cultivars Saehan (1.43 ± 0.14), Autumn Sense (1.19 ± 0.19), and Daesung (1.05 ± 0.16) demonstrated a 1.64-, 1.15-, and 1.34-fold higher reducing power in their unripe than their ripe fruits. In evaluating the reducing power of various fruit extracts, the studies conducted by Bae et al. [58] and Jeong et al. [59] showed that strawberry extract had a reducing power range from 0.37 to 1.10 at OD_700_; furthermore, the blueberry (1.98 at OD_700_) and the raspberry (1.59 at OD_700_) displayed higher reducing powers compared to kiwiberry cultivars. Additional research by Chung et al. [60] indicated that red kiwi (*Actinidia chinensis*) had a reducing power between 0.56 and 1.00, while Jeong [38] reported a range from 0.27 to 3.21 for gold kiwi. This variation in reducing power between the different fruit developmental stages of these kiwiberry cultivars highlights the complexity of their antioxidant properties and potential uses in functional foods.

In the present study, the kiwiberry extracts from various tissues of 12 cultivars demonstrated higher concentrations of phytochemicals and greater antioxidant capacities compared to other medicinal plants and previously studied kiwiberry species [25,35,36,39,40,58,59,60,83,84]. This was probably due to the genetically improved kiwiberry cultivars used in this study, compared to other kiwiberry species, or different weather and growth conditions that may have affected the antioxidant capacity. In the sample harvest year 2023, summer temperatures were on average 0.9 °C higher than the long-term mean recorded in 2013~2022. The average peak temperature in the summer period (June to August) reached 34.3 °C, which was significantly higher than the average high temperature of 29.4 °C observed in the previous decade [88]. The high dynamics of the bioactive compound metabolism are related to their function in the plants in signaling and protecting the compound metabolism, regardless of the genetic factor, in response to abiotic stresses [89,90,91]. Future studies should aim to explore the mechanistic links between environmental factors, like temperature, and antioxidant capacities, as well as the long-term effects of climate change on kiwiberry cultivars.

This study on phytochemical components, such as TPC and TFC, and antioxidant analysis of vegetative tissue and fruit extracts from kiwiberry cultivars noted limitations in the spectrophotometric methods used, including the Folin–Ciocalteu (F-C) method, which, while not perfect, is commonly employed for quantifying total phenolics. The spectrophotometric assay including F-C can sometimes interfere with the results because the other oxidation substrates present in the tested sample extract can interfere with the measurement of total phenolics in an inhibitory, additive, or enhancing manner [92]. Also, the possibility of losing free radicals during the ethanol extraction process in our study necessitates further consideration to ensure a more accurate representation of the antioxidant properties of the samples [93]. Despite these limitations, the spectrophotometric assays are simple, reproducible, and widely used for assessing phytochemical compounds and antioxidant activities in numerous tissues and cultivars. For more detailed profiling and accuracy, future studies plan to use high-performance liquid chromatography (HPLC) and electron paramagnetic resonance (EPR) techniques to assess a specific group and individual bioactive compounds in a more accurate manner in specific tissues of kiwiberry cultivars. 

## 3. Materials and Methods

### 3.1. Plant Materials and Growth Conditions

Various tissues of leaves, young stems, roots, and fruits from 12 kiwiberry (*A. arguta*) cultivars, i.e., cv. Gwangsan, cv. Green Ball, cv. Green Heart, cv. Daebo, cv. Daesung, cv. Saehan, cv. Apple, cv. Autumn Sense, cv. Wangneujdarae, cv. Cheongsan, cv. Cheongyeon, and cv. Chiak, were used in this study. Tested samples were collected from 10-year-old female kiwiberry trees in 2023 from a commercial kiwiberry growing orchard (coordinates: 37°27′06″ N, 127°53′18″ E; altitude: 127 m above sea level) in the city of Wonju, Gangwon, South Korea, characterized by a Dwa climate in Köppen climate classification [94,95]. In the plantation, one male tree is planted for every 10 female trees. The kiwiberry trees are planted in a 5 × 5 m spacing in 2013 and trained on a T-bar support. A compound fertilizer with an N:P:K ratio of 46.7:37.8:37.8 kg/ha is applied annually, and irrigation is performed twice a week, up to 15 mm per application, depending on soil moisture levels. No additional irrigation is provided during the rainy season from May to August. Compared to the long-term average (2013–2022), the mean monthly temperatures during the examined growing seasons (March–September) were 18.8 °C, while in the sample harvest year of 2023, it was 19.5 °C, increasing by 0.7 °C [88]. Consequently, the year 2023 was noticeably warmer than the long-term average temperatures. Fully expanded leaves, young stems, and roots were harvested in August. Unripe and ripe fruits were collected at 60 and 90 DAP, respectively. The ripe fruits (90 DAP) were harvested and allowed to ripen fully for 7 days at room temperature. The experiment consisted of three replicates, each with five female kiwiberry trees positioned similarly in relation to the male trees. From each of the five female trees per replicate, ten fully expanded leaves, stems, and roots, as well as 30 fruits, were randomly collected. All samples were immediately dried at 60 °C [25] or stored at −80 °C until use.

### 3.2. Kiwiberry Leaf and Fruit Biomass Analysis

To quantify leaf area, fully expanded detached leaves of the 12 kiwiberry cultivars were photographed, and the leaf area was measured using ImageJ 1.8.0 software (http://imagej.nih.gov/ij/). The FWs of the leaves and fruits were measured using an electronic scale (SPX2202KR, OHAUS, Parsippany, NJ, USA) immediately after harvest. After measuring FW, samples were fully dehydrated at 60 °C for 3 days and DWs were measured. The WC of leaves and fruits was calculated using the following equation: WC = [(FW − DW)/FW] × 100.

### 3.3. Kiwiberry Extracts

Plant extracts were collected from various kiwiberry tissues after the drying process, as previously described [96]. Briefly, 0.5 g of the dried samples was incubated with 25 mL of 99% ethanol (Daejung, Siheung, Korea) at 58 °C for 24 h using a shaking incubator (ED-SI300R, HYSC, Daejung), and the supernatants were collected after centrifugation (Allegra X-30R Centrifuge; Beckman Coulter, San Jose, CA, USA) at 1300× *g* for 15 min. The ethanol extracts from the kiwiberries were then used to measure the antioxidant activity.

### 3.4. Total Phenolic Content

TPC was determined using the FC method as described by Slinkard and Singleton [97]. Briefly, 50 μL of kiwiberry ethanol extract was combined with 50 μL of Folin–Ciocalteu’s phenol reagent (Sigma-Aldrich, St. Louis, MO, USA) and 250 μL of distilled water, and incubated for 5 min in the dark at room temperature (22–25 °C). Subsequently, 500 μL of 7% (*w*/*v*) sodium carbonate solution (Daejung) and 250 μL of distilled water were added to the mixture, which was incubated for 90 min. The absorbance of the mixture was measured at 760 nm using a multi-mode microplate reader (BioTek Synergy HTX; Agilent Technologies, Santa Clara, CA, USA). A standard curve used to quantify TPC was generated using gallic acid (Daejung). All assays were conducted in triplicate, and the results are presented as mean values. 

### 3.5. Total Flavonoid Content

TFC was measured using a method previously described by Zhishen et al. [98], with some modifications. In brief, 100 μL of kiwiberry extract was combined with 100 μL of 5% sodium nitrite solution (Daejung) and 50 μL of distilled water. This mixture was incubated at room temperature for 6 min, and then 150 μL of 10% (*w*/*v*) aluminum chloride solution (Daejung) was added, followed by a further 5 min of incubation. Subsequently, 200 μL of 1 M sodium hydroxide solution (Daejung) was added, and the mixture was incubated for 1 h at 37 °C. The absorbance of the solution was measured at 510 nm using a multi-mode microplate reader. A TFC standard curve was generated using quercetin (Sigma-Aldrich).

### 3.6. Free Radical Scavenging (DPPH) Activity

Free radical scavenging activity was determined using the DPPH based on the method described by Brand-Williams et al. [83], with slight modifications. A mixture comprising 100 μL of kiwiberry extract and 900 μL of 0.1 mM DPPH solution (Sigma-Aldrich) was incubated in the dark at room temperature for 15 min. The absorbance of the solution was measured at 515 nm using a multi-mode microplate reader. Gallic acid was used as a standard. The DPPH scavenging activity was calculated using the following equation:Inhibition (%) = (1 − sample absorbance/control absorbance) × 100

### 3.7. Nitrite Scavenging Activity

Nitrite scavenging activity (NSA) was determined using a method previously described by Kato et al. [84]. A mixture consisting of 50 μL of kiwiberry extract, 50 μL of 1 mM sodium nitrite solution (Daejung), and 300 μL of 0.1 N HCl (pH 1.2) was incubated for 1 h at 37 °C. Subsequently, 1 mL of 2% acetic acid (Sigma-Aldrich) solution and 100 μL of Griess reagent (Sigma-Aldrich) were added. After incubation for 15 min at room temperature, the absorbance of the solution was measured at 520 nm using a multi-mode microplate reader. NSA was calculated using the following formula:NSA (%) = [1 − (A − C)/B] × 100
where A is the absorbance of the sodium nitrite solution containing the plant extract and Griess reagent, B is the absorbance of the control (sodium nitrite solution with Griess reagent), and C is the absorbance of the sodium nitrite solution containing the plant extract and distilled water.

### 3.8. Reducing Power Assay

The reducing power assay was performed as described by Oyaizu [99]. A mixture of 300 μL of kiwiberry extract, 300 μL of 200 mM sodium phosphate buffer (pH 6.6), and 300 μL of 1% (*w*/*v*) potassium ferricyanide (Sigma-Aldrich) was incubated for 20 min at 50 °C in a water bath (JSWB-22(T); JSR, Korea). The reactants were mixed with 300 μL of 10% (*w*/*v*) trichloroacetic acid (Sigma-Aldrich) solution and then centrifuged at 13,000× *g* using a microcentrifuge (microfuge 20R; Beckman Coulter). Subsequently, 300 μL of 0.1% (*w*/*v*) ferric chloride (Sigma-Aldrich) was added to 300 μL of the supernatant, and the absorbance of the solution was measured at 700 nm using a multi-mode microplate reader.

### 3.9. ABTS Radical Cation Scavenging Activity

The ABTS [2,2′-azino-bis (3-ethylbenzothiazoline-6-sulfonic acid)] radical cation scavenging activity was measured using the method described by Lee and Kim [100]. Equal volumes of ABTS (Roche, Basel, Switzerland) solution and 2.6 mM potassium persulfate (Sigma-Aldrich) were mixed and incubated for 15 h in the dark. Afterward, the ABTS solution was diluted with distilled water to obtain an absorbance value of 0.7 ± 0.03 at 734 nm. Then, 60 μL of plant extracts and 900 μL of ABTS solution were mixed and incubated for 20 min in the dark. The radical scavenging activity was determined by measuring the absorbance at 734 nm using a multi-mode microplate reader. The ABTS scavenging activity (%) was calculated using the following formula: ABTS scavenging (%) = (1 − sample absorbance/control absorbance) × 100

### 3.10. Brix Analysis

Fresh kiwiberry fruits were used to extract juice and determine sugar content. Sugar concentrations were measured in °Bx units, representing the relative sugar content of the juice. These measurements were conducted in triplicate using a refractometer (Handheld Refractometer MyBrix, 30693200; Mettler Toledo, Columbus, OH, USA).

### 3.11. Total Sugar Content

The total sugar content was determined as described previously by DuBois et al. [101] and Lim et al. [102], with some modifications. A total of 0.5 g of kiwiberry fruits was extracted with 10 mL of 50% methanol (Daejung) at 80 °C for 30 min using a water bath (WCB-22, DAIHAN, Korea). The supernatant was collected from the fruit extract using a centrifuge (Allegra X-30R Centrifuge; Beckman Coulter) at 1000× *g* for 10 min, and 25 μL of kiwiberry extract and 25 μL of 5% phenol (Daejung) were mixed and vortexed for 30 s. Subsequently, samples were placed on ice, 125 μL of sulfuric acid (Daejung) was added to each sample, and they were incubated at 80 °C for 30 min. After cooling, the absorbance was measured at 490 nm using a multi-mode microplate reader (BioTek Synergy HTX; Agilent Technologies). A standard curve was generated using different concentrations of glucose as the standard for quantification of the total sugar content. 

### 3.12. Vitamin C

Vitamin C content was measured using the method described by Kang and Song [103]. Briefly, 5 mL of 5% metaphosphoric acid solution (Daejung) was added to 2.5 g of kiwiberry fruit sample and vortexed for 5 min. The mixture was centrifuged at 1000× *g* for 10 min, and the clear supernatant was collected for analysis. A 1 mL aliquot of the supernatant was mixed sequentially with 25 μL of 2, 6-dichlorophenol-indophenol (Fisher Scientific, USA), 1 mL of 2% thiourea (Daejung), and 1 mL of 2, 4-dinitrophenylhydrazine (Sigma), and the reaction mixture was then incubated at 37 °C for 3 h. The mixture was immediately cooled on ice, and 85% sulfuric acid (2.5 mL) was carefully added. The samples were incubated at room temperature for 30 min, and the absorbance was measured at 540 nm using a microplate reader. A vitamin C standard curve was generated using different concentrations of ascorbic acid dissolved in a 5% metaphosphoric acid solution.

### 3.13. Statistical Analysis

All of the standard statistical analyses were conducted using Prism9 software (https://www.graphpad.com). Data are expressed as mean ± standard deviation of the values measured using three independent biological replicates for all extracts, with two technical replicates for each kiwiberry tissue. One-way analysis of variance (ANOVA) was performed to test for significance differences in the measurements between cultivars and to compute the adjusted *p*-values and level of significance, followed by Tukey’s multiple comparison test [104]. Data were evaluated and marked as significantly different when *p*-values were below 0.05.

## 4. Conclusions

The domestication of the native kiwiberry (*Actinidia arguta*) in the last century marks a significant advancement in agricultural practices, drawing attention to its diverse nutritional benefits for humans. This study on the phytochemical content and antioxidant activity of 12 kiwiberry cultivars contributes to the development of novel functional biomaterials based on its vegetative tissues and fruits. Specifically, root tissues of most kiwiberry cultivars are particularly notable for their high antioxidant activity, offering significant potential for various applications. The TPC, DPPH, ABTS, and reducing power of leaf extracts from cv. Daebo were 2.8, 2.5, 3.6, and 2.9-fold higher than those of other kiwiberry cultivars. In the stems, the DPPH and ABTS activity of kiwiberry cv. Saehan and cv. Autumn Sense were 4.2-fold higher than in the other cultivars. The TPC, TFC, and reducing power of root extracts from kiwiberry cv. Saehan were 3.4, 1.5, and 2.5-fold higher, respectively, than in the other cultivars. The kiwiberry cultivars Saehan and Daebo, with their rich antioxidant properties in both the vegetative tissues and fruits, along with their high biomass, can be utilized as valuable resources for applications in health foods, beverages, and the cosmetic industry. Further studies will be conducted on the identification of phytochemical and antioxidant compounds, combined with the study of their biosynthetic pathways and the molecular mechanism of the derived bioactivity. 

## Figures and Tables

**Figure 1 ijms-25-01505-f001:**
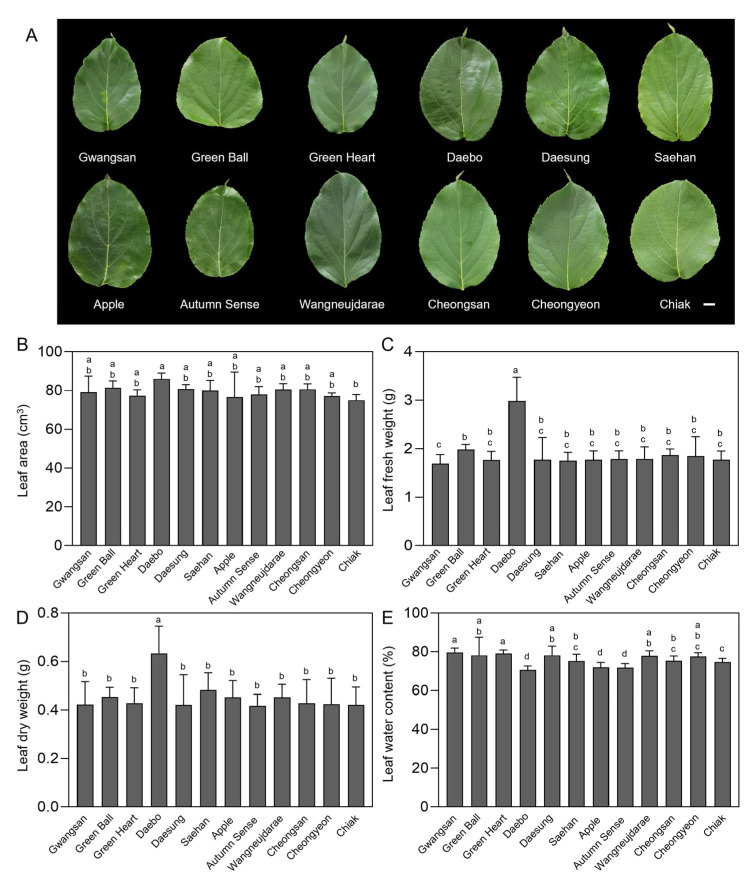
Comparative analysis of leaf morphology and biomass components in kiwiberry cultivars. Representative images of fully expanded leaf of each of the 12 kiwiberry cultivars. Scale bars = 1 cm (**A**). Fully expanded leaves from each of the 12 kiwiberry cultivars were evaluated for their leaf area (**B**), leaf fresh weight (**C**), leaf dry weight (**D**), and leaf water content (**E**). The values represent the mean ± standard deviation (*n* = 3 biological replicates, with 10 detached leaves per replicate). Values marked with a different letter on each chart were considered significant at *p* < 0.05 (Tukey’s multiple comparison test).

**Figure 2 ijms-25-01505-f002:**
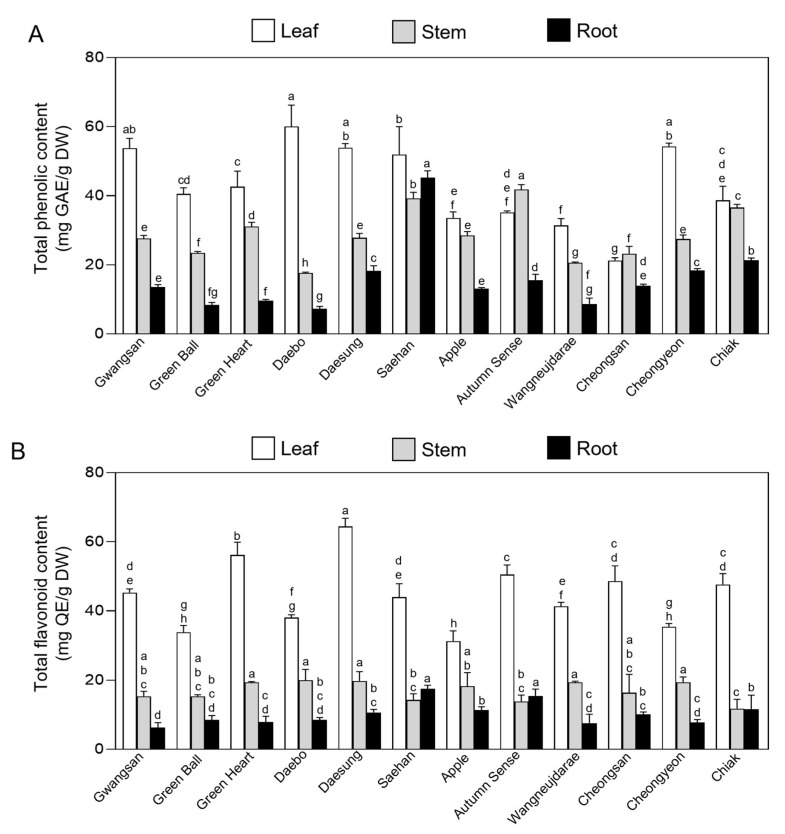
Phytochemical composition of the leaf, stem, and root extracts from each of the 12 kiwiberry cultivars. Total phenolic content (**A**) and flavonoid content (**B**) were measured in different kiwiberry cultivars. The values represent the mean ± standard deviation (*n* = 3 biological replicates, with 10 detached leaves, stems, and roots per replicate). Values marked with a different letter on each chart were considered significant at *p* < 0.05 (Tukey’s multiple comparison test).

**Figure 3 ijms-25-01505-f003:**
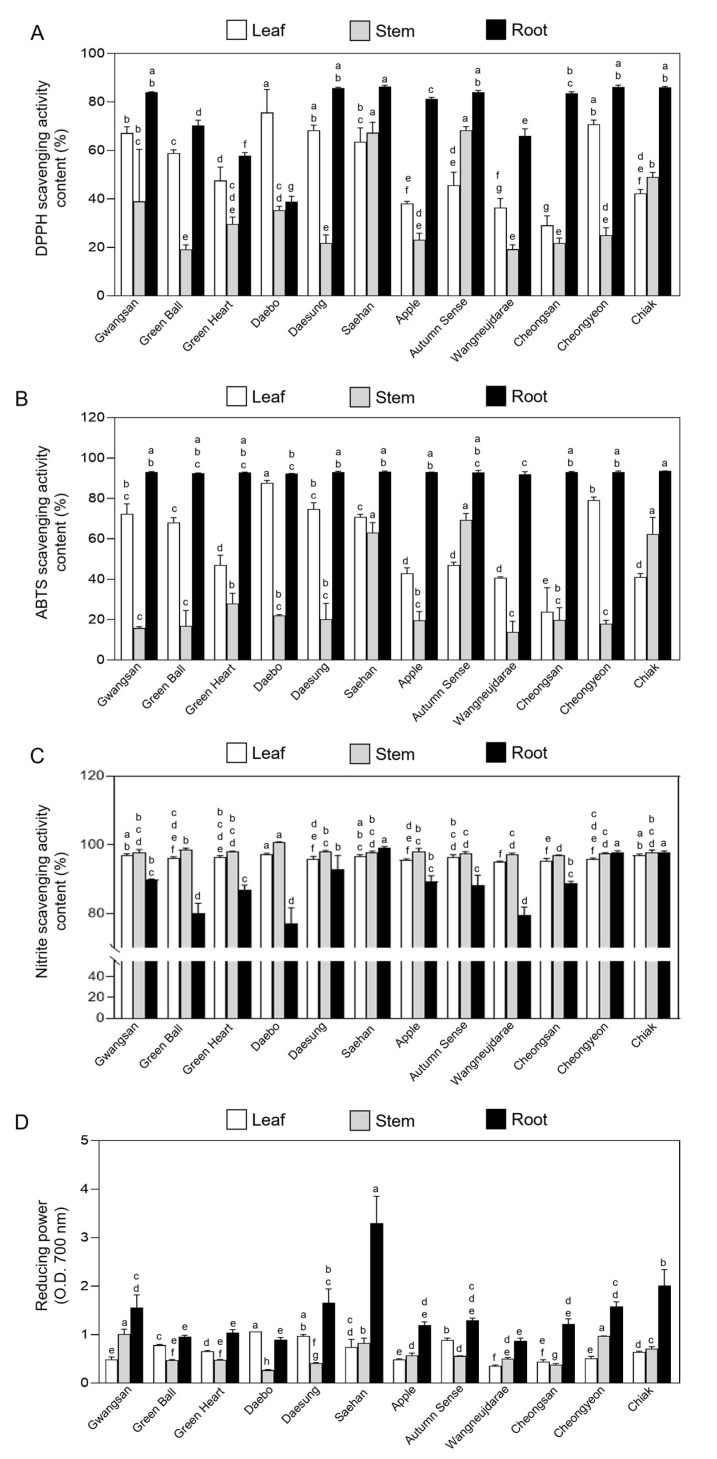
Antioxidant activity of the leaf, stem, and root extracts from each of the 12 kiwiberry cultivars. The DPPH radical scavenging activity (**A**), ABTS radical scavenging activity (**B**), nitrite scavenging activity (**C**), and reducing power activity (**D**) were measured in each kiwiberry cultivar The values represent the mean ± standard deviation (*n* = 3 biological replicates, with 10 detached leaves, stems, and roots per replicate). Values marked with a different letter on each chart were considered significant at *p* < 0.05 (Tukey’s multiple comparison test).

**Figure 4 ijms-25-01505-f004:**
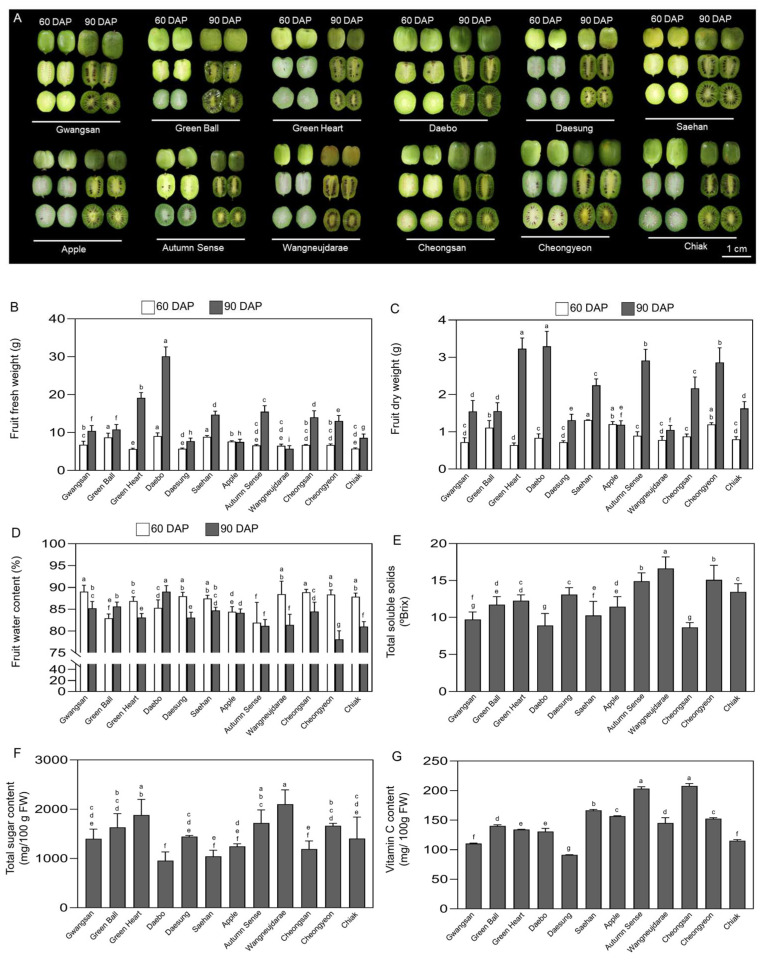
Comparative analysis of fruit morphology (**A**), fresh weight (**B**), dry weight (**C**), water content (**D**), sugar content in Brix (**E**), total sugar content in mg 100 g^−1^ FW (**F**), and vitamin C content (**G**) were measured in different kiwiberry cultivars. Representative fruit images of each of 12 kiwiberry cultivars were captured at 60 and 90 days after pollination (DAP). Scale bars = 1 cm. The values represent the mean ± standard deviation (*n* = 3 biological replicates, with 30 detached kiwiberry fruits per replicate). Values marked with a different letter on each chart were considered significant at *p* < 0.05 (Tukey’s multiple comparison test).

**Figure 5 ijms-25-01505-f005:**
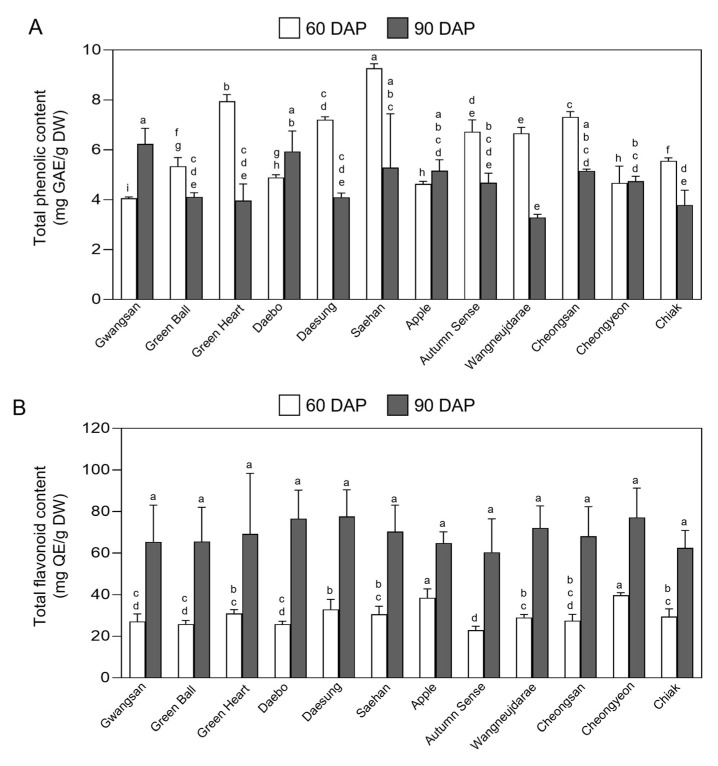
Phytochemical composition of fruit extracts from each of the 12 kiwiberry cultivars. Total phenolic content (**A**) and flavonoid content (**B**) of unripe (60 DAP) and ripe (90 DAP) fruit extracts were measured in different kiwiberry cultivars. The values represent the mean ± standard deviation (n = 3 biological replicates, with 30 detached kiwiberry fruits per replicate). Values marked with a different letter on each chart were considered significant at *p* < 0.05 (Tukey’s multiple comparison test).

**Figure 6 ijms-25-01505-f006:**
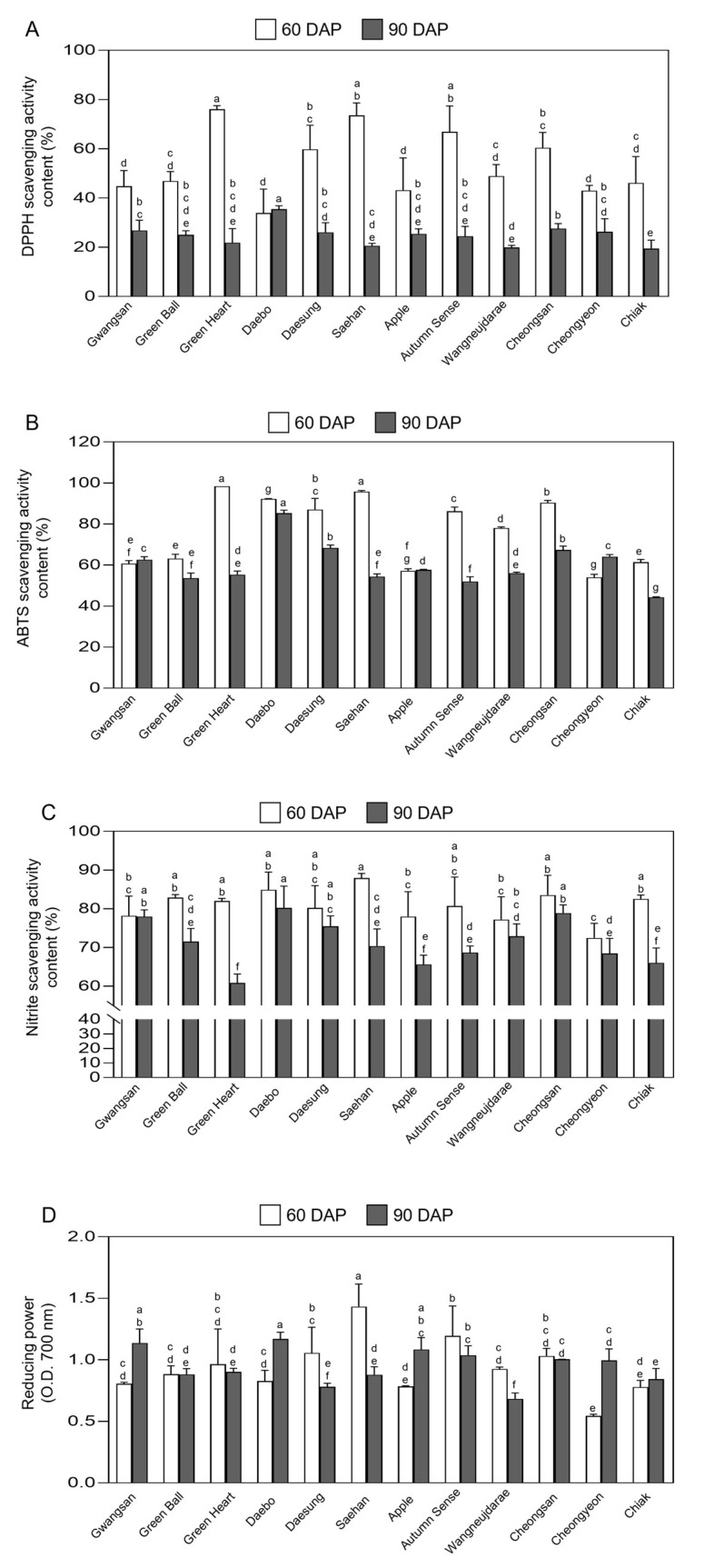
Antioxidant activity of fruit extracts from each of 12 kiwiberry cultivars. The DPPH radical scavenging activity (**A**), ABTS radical scavenging activity (**B**), nitrite scavenging activity (**C**), and reducing power activity (**D**) of unripe (60 DAP) and ripe (90 DAP) fruit extracts were measured in different kiwiberry cultivars. The values represent the mean ± standard deviation (*n* = 3 biological replicates, with 30 detached kiwiberry fruits per replicate). Values marked with a different letter on each chart were considered significant at *p* < 0.05 (Tukey’s multiple comparison test).

## Data Availability

All data are available within the manuscript.

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
