# Peer review of "Antioxidant Activity Analysis of Native Actinidia arguta Cultivars"

_ijms, 2024, doi:10.3390/ijms25031505_

Round 1
Reviewer 1 Report
Comments and Suggestions for Authors
This study aimed to investigate the antioxidant activity of 12 kiwi berry cultivars using the TPC, TFC, DPPH, and ABTS radical scavenging activity, nitrite scavenging 18 activity, and reducing power in leaves, stems, roots, and fruits. There is no doubt that all data obtained are valuable. Moreover, all experiments were performed, and the results were presented well. On the other hand, more is needed to be published in the IJMS journal:
1. What is actually presented in Figure 2 is the total phenolic and flavonoid contents of the samples. It would be more accurate to write 'phytochemical composition' instead of 'antioxidant activity' in the figure title. The same applies to Figure 5.
2. Although root extract was the richest extract in terms of phenolics and flavonoids in phytochemical composition analyses, the reasons for this extract's high radical scavenging activity should be discussed convincingly.
3. Is it really necessary to provide detailed information about leaf and/or fruit morphologies when analyzing the antioxidant activities of kiwiberry cultivars?
4. Analyzing the chemical composition of samples only by qualitative tests would mean repeating the generalization that phenolics/flavonoids contribute to antioxidant activity. It would be more accurate to use quantitative chromatographic techniques to detect the chemicals responsible for the activity.
Reviewer 2 Report
Comments and Suggestions for Authors
I have gone through the manuscript with great interest. The study seems noteworthy due to its significance in identifying the biochemical attributes of native Actinidia argute (Kiwifruit) of Korea. Overall, this study provides valuable insights into the antioxidant activity of various kiwiberry cultivars, shedding light on their phenolic content and potential as functional foods. The use of different parameters such as TPC, TFC, DPPH and ABTS radical scavenging activity, nitrite scavenging activity, and reducing power in different plant parts (leaves, stems, roots, and fruits) adds depth to the analysis. This may contribute to sustainable kiwifruit production in the specific region.
Overall, the manuscript is well-written, and the study is promising and contributes valuable information to the field. However, the following suggestions are offered to enhance the manuscript's utility.
1. Abstract:
· Provide a brief description of the experimental site location, agro-climatic conditions, and agronomic practices.
· Mention the statistical analysis used in the study.
· Provide the significance/objective of the research in this part.
· Please recommend/ suggest at least two elite cultivars
2. Introduction:
· Strengthen the written English in this section.
· Add the worldwide production of Kiwi fruit, also mention its production and consumption in Korea.
· Minimize the botany description in first two paragraphs. Please focus and discuss the impact of climate on kiwi fruit attributes.
· Need to clarify research objectives.
3. Materials and Methods:
· Elaborate on the uniformity of agronomic management practices and plant health status.
· Leaf biomass parameters, including leaf area, fresh weight (FW), 78 dry weight (DW), and water content (WC) were discussed in the results and discussion part. However, the methodology has not been added in this part. Please add complete details of each parameter in this part, how they were calculated or measured?
4. Results and Discussion
While the manuscript presents this section well, consider including information about agronomic practices for a more comprehensive discussion.
· Strengthen the argumentation with relevant citations and results.
Comments on the Quality of English LanguageIt can be improved by reviewing through English native
Reviewer 3 Report
Comments and Suggestions for Authors
This study was conducted to investigate the antioxidant activity in leaves, stems, roots, and fruits of 12 kiwiberry cultivars. Total polyphenol (TPC), total flavonoid (TFC), DPPH and ABTS radical scavenging activity, nitrite scavenging activity, and reducing power on kiwiberry extracts was determined. Based on their results, the authors conclude that kiwiberry has significant phenolic content and antioxidant activity, thus having potential as a functional food and natural antioxidant. In particular, the authors underline that, even if numerous studies concern the antioxidant properties of the kiwiberry fruit, those concerning leaves, stems, and roots tissues in kiwiberry cultivars have not been extensively studied. The novelty of this study is therefore the comparison of the antioxidant properties of various kiwiberry tissues.
As a matter of facts, in this reviewer’s opinion, many parameters were determined in this study: among them, the nitrite and free radical scavenging activity, the reducing power, as well as the total sugar and the Vitamin C contents. However, data presentation and discussion need a deep major revision in the following points:
1) First, the number of samples for each cultivar used for the experiments. In Materials Section we find the nonspecific expression ’Various’ tissues, while the sample number is never reported there. However, in The Caption of the Figures ‘n=3’ can be found. Is it the numerosity? If this is the number of the experimental samples, this must be clearly reported, but even more, the authors must demonstrate that n=3 is a number sufficient for the significance of the data. Otherwise, the whole study is a non-sense. To this aim, the Prospective calculation of the power to determine significant numbers must be made by using the Freeware G*Power software (http://www.psycho.uni-duesseldorf.de/abteilungen/aap/gpower3/) At a power of 80%, the calculated number of significant subjects must result equal or lower than the number of samples used, otherwise, as highlighted below, the results are meaningless.
2) In this study, the antioxidant properties of the kiwi tissues and fruits were mainly determined on ethanol extracts. How can the authors assure that free radicals are not lost during the procedure to obtain the samples? The authors didn’t address this problem. A direct measure for comparison is suggested. This can be easily obtained by the Electron Paramagnetic Resonance technique.
3) In a scientific manuscript all data must be reported as mean values and SD. A range and/or a simple percent without SD was incorrectly used in this manuscript.
4) The manuscript is extremely poorly written and needs to be largely reworked. The organization is not clear; the acronyms are carelessly defined; the arguments are hard to follow. Results and Discussion presented in a unique section, even more missing a Conclusion section, makes the manuscript confusing. General sentences must be moved in the Introduction. Significant correlations miss the significance level (see for example line 232) and highest and/or lowest expressions don’t belong to a scientific report.
5) The authors must clearly present their results, in particular the differences between kiwi tissues and kiwi fruits must be emphasized.
6) The limitations of the study must be reported, preferably in a separate section.
7) An abbreviation list is recommended.
Comments on the Quality of English Language
Extensive editing required
Reviewer 4 Report
Comments and Suggestions for Authors
Dear authors this study provides new and important data about the use Actinidia arguta cultivars as an antioxidant. It is a very important study because 12 from a total of 75 species of kiwiberry (Actinidia arguta) cultivars were investigated for their antioxidant properties. I also must propose the investigation of the antioxidant properties of these cultivars on animal models such as broilers or layers to evaluate how these antioxidant properties affect the meat and egg products respectively. I strongly recommend this study be published in IJMS without any revision.
Reviewer 5 Report
Comments and Suggestions for Authors
This paper, entitled Antioxidant activity analysis of native Actinidia arguta cultivars, is a scholarly work and can increase knowledge on this domain. The authors provide an interesting and original study, the content is relevant to Intenational Journal of Molecular Sciences.
I have some general and specific comments:
- The abstract and keywords are meaningful.
- The manuscript is quite well written and well related to existing literature.
- Please enlarge or resize the Figures in order to improve the readability.
As it, the paper is fully acceptable for publication but requires minor amendment especially considering readability and size of the figures. I recommend the follwing decision: ACCEPT AFTER MINOR REVISION.
-
Round 2
Reviewer 1 Report
Comments and Suggestions for Authors
The manuscript has been revised according to the comments, but the quantitative experiments have yet to be conducted.
Reviewer 3 Report
Comments and Suggestions for Authors
I consider the manuscript in its revised version suitable for publication.
Comments on the Quality of English LanguageMinor editing required